# Seroprevalence of brucellosis in small ruminants and related risk behaviours among humans in different husbandry systems in Mali

Souleymane Traoré[1]*, Richard B. Yapi[2,3], Kadiatou Coulibaly[4], Coletha Mathew[5], Gilbert Fokou[2], Rudovick R. Kazwala[5], Bassirou Bonfoh[2], Rianatou Bada Alambedji[1]

1 Ecole Inter-Etats des Sciences et Médecine Vétérinaires de Dakar, Dakar, Sénégal, 2 Centre Suisse de Recherches Scientifiques en Côte d'Ivoire, Abidjan, Côte d'Ivoire, 3 Centre d'Entomologie Médicale et Vétérinaire, Université Alassane Ouattara, Bouaké, Côte d'Ivoire, 4 Laboratoire Centrale Vétérinaire de Bamako, Bamako, Mali, 5 Sokoine University of Agriculture, Morogoro, Tanzania

* souleymanot@yahoo.fr

**Data Availability Statement:** All relevant data are within the paper and its Supporting information files.

## Abstract

Mali has a high pastoral potential with diverse coexisting production systems ranging from traditional (nomadic, transhumant, sedentary) to commercial (fattening and dairy production) production systems. Each of those systems is characterised by close interactions between animals and humans, increasing the potential risk of transmission of zoonotic diseases. The nature of contact network suggests that the risks may vary according to species, production systems and behaviors. However, the study of the link between small ruminants and zoonotic diseases has received limited attention in Mali. The objective of this study was to assess brucellosis seroprevalence and determine how the husbandry systems and human behaviour expose animal and human to infection risk. A cross-sectional study using cluster sampling was conducted in three regions in Mali. Blood was collected from 860 small ruminants. The sera obtained were analysed using both Rose Bengal and cELISA tests. In addition, 119 farmers were interviewed using a structured questionnaire in order to identify the characteristics of farms as well as the risk behaviors of respondents. Husbandry systems were dominated by agro-pastoral systems followed by pastoral systems. The commercial farms (peri-urban and urban) represent a small proportion. Small ruminant individual seroprevalence was 4.1% [2.8–5.6% (95% CI)]. Herd seroprevalence was estimated at 25.2% [17.7–33.9% (95% CI)]. Peri-urban farming system was more affected with seroprevalence of 38.1% [18.1–61.5 (95% CI)], followed by pastoral farming system (24.3% [11.7–41.2 (95% CI)]). Identified risk behaviors of brucellosis transmission to animals were: exchange of reproductive males (30.2%); improper disposal of placentas in the farms (31.1%); and keeping aborted females in the herd (69.7%). For humans, risk factors were: close and prolonged contact with animals (51.2%); consumption of unpasteurized dairy products (26.9%); and assisting female animals during delivery without any protection (40.3%). This study observed a high seroprevalence of brucellosis in small ruminants and also identified risky practices

**Funding:** The authors acknowledge support from the DELTAS Africa Initiative [Afrique One-ASPIRE /DEL-15-008, http://afriqueoneaspire.org/)]. Afrique One-ASPIRE is funded by a consortium of donor including the African Academy of Sciences (AAS) Alliance for Accelerating Excellence in Science in Africa (AESA), the New Partnership for Africa's Development Planning and Coordinating (NEPAD) Agency, the Wellcome Trust [107753/A/ 15/Z] and the UK government. The funders had no role in study design, data collection and analysis, decision to publish, or preparation of the manuscript.

**Competing interests:** The authors have declared that no competing interests exist.

that allow cross transmission between the two populations. This calls for control strategy using a multi-sectoral and multidimensional approach.

## Introduction

Livestock plays an important economic and sociocultural role in Mali with its contribution to gross domestic product estimated at 13% [1]. More than 80% of the rural population are involved in animal husbandry [2]. The national ruminants population is estimated at 11,415,900 cattle, 17,400,000 sheep, 24,023,800 goats and 1,192,900 camels [2]. The various husbandry systems are classified as traditional (sedentary, pastoral and agropastoral) and commercial (fattening and milk production) [3,4]. In general four main systems are identified. Firstly, there are agropastoral systems where agriculture and livestock coexist. Depending on the areas and the predominance of agriculture or livestock, there are two subsystems [3,5]. Secondly, the pastoral systems are subject to climatic constraints and characterized by the great mobility of herds related to availability of natural resources [4,6]. Thirdly, peri-urban systems are generally characterised by dairy farming and small ruminants keeping. The animals graze within half a day on the farm [6]. The urban systems are composed of small farms (less than 10 head) inside the families in the cities. In general, these are farms with products that are not for commercial purposes [6]. Livestock sector in Mali is facing many health concerns including diseases that hinder production and development of the sector in the country. There are many diseases facing livestock sector, including brucellosis [3]. Brucellosis remains the most common zoonotic infection worldwide, and is caused by a bacteria of the genus *Brucella* [7]. The species *B. melitensis* which is the most widespread in small ruminants and known to be the most pathogenic in humans [8,9]. Studies around the world have linked human seroprevalence to that of animals [10,11]. The seroprevalence of brucellosis in livestock varies according to husbandry systems [12]. Some studies have highlighted that husbandry systems might influence animal seropositivity [12,13], whilst others find no link between these systems and the infection [14,15]. In Mali, the first cases of human brucellosis were observed in 1939 by Sicé in the circle of Gao on two patients [16]. From that time period, most of the investigations conducted so far were in cattle [17–19]. Only two studies were conducted on small ruminant which reported a seroprevalence of 0.7% in the municipality of Cinzana [20] and 37.1% in Niono [21]. These two studies were conducted in the Segou region alone and on relatively small sites. Furthermore the two studies did not consider the diversity of husbandry systems in their study design. To address this knowledge gap, our study was undertaken to determine the role of small ruminant husbandry systems in maintaining and transmitting brucellosis in the country. Specifically, the study aims to assess the seroprevalence of brucellosis and show how the husbandry system and the human behaviours expose animals and human to the disease.

## Methods

### Study area

This study was conducted in Mali and precisely in the regions of Sikasso, Segou and Bamako District. These regions were chosen on the basis of their difference in agroclimatic conditions and animal husbandry practices. The selected area represents 32% of the total number of small ruminants in Mali. The climate of Sikasso is Sudanian tropical type, subdivided into two climatic groups: the Sudanese wet zone and the Guinean zone. It is the most humid region of Mali and the most watered (annual rainfall ranging from 700 to 1500 mm). The predominant husbandry system in Sikasso is the agropastoral system. The region of Segou, located in central

Mali, has a semi-arid climate (average annual rainfall: 513 mm). The presence of several rivers (it is crossed by the Niger River (over 292 km) and the Bani River) allows irrigated crops. The main husbandry systems are pastoral and agropastoral. Like in Sikasso, the climate of Bamako area is Sudanian type. It has a humid tropical climate with a total annual rainfall of 878 millimetres. Due to the proximity the urban centre, this area has commercial husbandry system (fattening and dairy production). Field data was collected from 5[th] November to 12[th] December 2018.

## Study design and sample size calculation

This was a cross-sectional study design using cluster sampling method. The following formula was used to calculate the sample size:

$$n = [D * Np(1 - p)]/[(d^2/Z^2_{1-\alpha/2} * (N - 1) + p * (1 - p)].$$

Where n = sample size; D = design effect; p = expected seroprevalence; d = precision; $Z_{1-\alpha/2}$ = 1.96.

The sample size calculation was based on the expected seroprevalence of brucellosis of 37.1% in small ruminants reported by Coulibaly in 2016 [21]. The design effect was calculated using following formula: D = 1 + (b—1) ρ [22]; where b = average number of animal by cluster, ρ = intra cluster correlation coefficient, for ρ, we used 0.2 for small ruminants brucellosis [23]. To target a 95% confidence interval between 0.320 and 0.420, we got D = 1.8. Finally, the minimum size necessary to consider for the study was 650 animals. To be included in this study, each farms had to meet the following criteria: (i) consent of the owner or breeder, (ii) a minimum size of 10 animals, (iii) animal to be sampled must be at least six months old.

## Sampling

We considered in the sampling method, the administrative (regions) and settings (villages and hamlets) in the selection of the study sites. In each region, sampled areas were selected randomly. Regarding the Bamako District, villages or hamlets were also selected randomly, considering the four axes of city as epidemiological units. In accordance of the sample size, the number of animals to be selected per farm and the localities in which study should take place, it was necessary to choose six (6) farms per locality (villages, hamlets) because our minimum sample size was 126, we had 21 localities and we wanted equal number of samples in each locality. In Sikasso region, nine municipalities were sampled, while eight municipalities were selected in Segou and the four main axes of Bamako were considered. This gave 54 farms in Sikasso, 48 farms in Segou and 24 farms in Bamako to be sampled. In each farm, at least five animals (sheep and goats) were selected randomly according to the herd structure. Afterward, one person (the owner or the person in charge of the farm) was interviewed using a standardised questionnaire regarding practices related to risk behaviors for brucellosis infection, for both human and animal.

## Data collection

**Questionnaire administration.** The structural and functional characteristics of the farms were obtained through a questionnaire and by direct observations. The questionnaire including questions on the main risk factors for the transmission and maintenance of *Brucella* infection in livestock and humans previously identified by other authors [11,24,25]. The questionnaire included also questions related to livestock health status, occurrence of abortions, herd proximity to permanent watering bodies, herd mixing practices. Individual animal

demographic data such as sex, age, history of abortion was also collected. Knowledge and risk factors related to brucellosis were assessed with participants through questions on the consumption patterns of animal products, obstetric practices, and exposure to livestock products. The socio-demographic characteristics (gender, age, profession) of the respondents were recorded and knowledge on symptoms of brucellosis in humans was captured. The questions were addressed to breeders and /or herd owners. The questionnaire was designed in French and pretested with 10 breeders in the outskirts of the district of Bamako. The survey lasted for about 10 minutes and was conducted in Bambara and/or in French. Written consent of the respondents was obtained by the principal investigator before each interview.

**Blood sample collection.**   Five millilitres of blood were collected from jugular vein in a plain tube identified with a unique code. After centrifugation at 5000rpm for 3 min, the sera were removed and put in cryotubes. The sera were stored in a freezer (-20˚C) at the Laboratoire Central Vétérinaire de Bamako before transportation to the Laboratoire de Microbiologie Immunologie et Pathologie Infectieuse de Dakar, for serological tests.

**Serological tests procedure.**   The competitive Enzyme Linked Immunosorbent Assay (cELISA) and Rose Bengal Test (RBT) were used for serological tests. Both tests were used in parallel. Adamou finds that the combination of the two tests gives a sensitivity of 97.2% and a specificity of 93.5% [26].

The RBT is a rapid, simple, economical test, with sensitivity (Se1 = 80.2%) and specificity (Sp1 = 99.7%) [27]. The competitive ELISA has sensitivity (Se2 = 98.8) and specificity (Sp2 = 97.6). These tests meet the requirements of the World Organisation for Animal Health (OIE). Briefly, for RBT, the Rose Bengal antigen was mixed with the serum and observed for about four minutes whrepresence of agglutination indicates a positive reaction.

The principle of cELISA is based on competition between the original antibodies (sample) and the conjugate added. According to the manufacturer's recommendations, the reactions should take place at a temperature of 21˚C ± 6. In the laboratory, the air conditioner was set at a temperature of 21˚C. A thermometer was used to monitor this temperature. After incubation, the antibody-antigen complex was formed. Following washing, unbound antibodies were removed. After the addition of the substrate, the remaining enzymes cause a chromogenic or fluorescent signal readable by the naked eye or by a chromatograph reader. The lack of colour development indicates that the sample tested was positive. A positive/negative cut-off was calculated as 60% of the mean of the optical density (OD) of the 4 conjugate control wells. Any test sample giving an OD equal to or below this value was considered as positive. Positive and negative controls were used to control the quality of the results obtained.

## Data analysis

The statistical analyses were performed with R software (RStudio-1.0.136). The different husbandry systems were identified by calculating the percentage of structural and functional characteristics of the farms. The farms were categorized based on the definitions given by previous authors [3–6]. Also, the structural and functional characteristics of the farms (size, enclosure, movement, etc.) were considered.

To determine the seroprevalence of brucellosis according to the husbandry system, the results of serological tests were converted into binary variables (positive = 1, negative = 0). A herd was considered positive if at least one animal in that herd was positive either to RBT or cELISA. Individual seroprevalence was calculated by the number of positive animals over the number of animals tested. Combined sensitivity and specificity of the two tests used in parallel was calculated using the following formulas: Sensitivity (Se) = 1 –(1 –Se1) * (1 –Se2) and specificity (Sp) = Sp1 * Sp2 [27]. Where se1: sensitivity of first test and se2: sensitivity of second test,

sp1: specificity of first test, specificity of second test. The different risk behaviors cited by the farmers were recorded. Only individual with complete data were included in the final analysis. The percentages and proportions of these risk behaviors were determined. Chi-square test was calculated, and it was deemed significant when p≤0.05.

### Risk factor analysis

Univariate analysis was conducted for explanatory variables (associated with brucellosis sero-positivity) and those with a *P* value ≤ 0.25 were taken into the multivariate logistic regression model. In the multivariate logistic regression model, selections were made until a model with explanatory variables having a *p* value ≤ 0.05 was obtained.

### Ethical considerations

This study obtained the approval of the ethics committee of the « Institut National de Recherche en Santé Publique (INRSP) du Mali » following the decision N˚ 23/2018 / EC-INRSP. In addition, permission to carry out the study was obtained from the livestock owners. An informed consent document detailing the research procedure was made available for them and a written informed consent was obtained. For minors, an acquiescence form was made available to them and parental consent was obtained. Blood sampling procedures of animals were done with respect for animal welfare [28].

## Results

### Descriptive analysis of the study population

A total of 860 small ruminants (514 sheep, 346 goats) were sampled. These animals belonged to 119 farms that were in 49 localities (villages and hamlets). Among the 119 farms surveyed, 52.1% were mixed (sheep and goat) herds, 33.6% and 14.3% were only composed of sheep and goats, respectively. Out of 119 farmers 78.1% owned other animal species in addition to small ruminants, such as cattle, poultry, equines and swine. The farmers had in their possession various breeds of small ruminants. These are Sahelian breed like Djallonke, the crossbreed from crossing of exotic breeds with local breeds and exotic breeds (Chadian sheep, Sudanese sheep, Guerra goats). The average size of a herd surveyed was 37.1 [30.8–43.4 (95% CI)]. Out of the 119 farms surveyed, 46.2% were from the Sikasso region, 30.2% from Segou and 23.2% from the periphery of Bamako District.

 Among the respondents, 87.3% were men and 12.6% were women, 45% were over 45 years old and 43% were between 25 and 45 years old. Of the 119 people interviewed, 70.6% were educated. Of these educated, 33% went to French school, 29.4% to Koranic school and 8% received instructions in local languages. Small ruminant breeders have other activities in parallel; there are traders, teachers, soldiers, veterinarians and retirees. According the responses, 76.5% of the animal owners constituted their herd by purchase, 20.1% by inheritance from their parents whilst, 3.3% constituted theirs from donations (Table 1).

### Husbandry systems

The farming systems identified were: Agropastoral, Pastoral, peri-urban and urban.

 Agropastoral system where agriculture and livestock coexist accounted for 43.7% [34.5–53.1% (95% CI)] of the farms investigated whilst 31.1% [22.9–40.2% (95% CI)] belonged to the pastoral system characterized by the great mobility of herds depending on the availability of natural resources. Peri-urban and urban systems characterised by semi-sedentary dairy farmers and small farms (less than 10 head) inside the concessions in the city. The semi-sedentary

**Table 1. Sociodemographic characteristics of study respondents.**

| Characteristics Category | Bamako [n (%)] | Segou [n (%)] | Sikasso [n (%)] | Total (%) |
|---|---|---|---|---|
| **Sex** | | | | |
| Female | 3 (2.53) | 3 (2.52) | 9 (7.5) | 12.6 |
| Male | 25 (21) | 33 (27.7) | 46 (38.6) | 87.3 |
| **Age** | | | | |
| [0 18] | 1 (0.8) | 0 (0.0) | 1 (0.8) | 1.6 |
| [18 25] | 7 (5.8) | 2 (1.6) | 4 (3.3) | 10.7 |
| [25 45] | 9 (7.6) | 15 (12.7) | 27 (22.7) | 43 |
| [over 45] | 11 (9.5) | 19 (16.0) | 23 (19.5) | 45 |
| **Education** | | | | |
| no | 7 (5.9) | 7(5.9) | 21 (17.6) | 29.4 |
| yes | 21 (17.6) | 29 (24.4) | 34 (28.6) | 70.6 |
| **Type of education attended** | | | | |
| Koranic | 7 (5.8) | 16 (13.5) | 12 (10.1) | 29.4 |
| French | 14 (11.8) | 8 (6.8) | 17 (14.4) | 33 |
| Local | 0 (0.0) | 4 (3.3) | 5 (4.2) | 7.5 |
| **Education level** | | | | |
| Basic | 8 (6.7) | 8 (6.7) | 11 (9.2) | 22.6 |
| Secondary | 7 (5.8) | 12 (10.0) | 12 (10.0) | 25.8 |
| High | 6 (5.0) | 9 (7.5) | 11 (9.2) | 21.7 |
| **Main occupation** | | | | |
| Agriculture | 3 (2.5) | 8 (6.7) | 23 (19.3) | 28.5 |
| Livestock keeping | 9 (7.5) | 14 (11.7) | 15 (12.6) | 31.8 |
| Other | 16 (13.4) | 14 (11.7) | 17 (14.2) | 39.7 |
| **Herd acquisition** | | | | |
| Purchase | 27 (22.6) | 25 (21.2) | 39 (32.7) | 76.5 |
| Donation | 1 (0.8) | 0 (0.0) | 3 (2.5) | 3.3 |
| Inheritance | 0 (0.0) | 11 (9.2) | 13 (10.9) | 20.1 |

Data stem from data collected from November 5 to December 12, 2018 in the region of Bamako, Segou and Sikasso, n = number of individuals, (%) = percentage.

dairy farmers and small farms represented 17.6% [11.2–25.7% (95% CI)] and 7.5% [3.5–13.8% (95% CI)] respectively of the farms surveyed. According to region, Segou was dominated by pastoral system whilst in Sikasso the agropastoral system was dominant. Peri-urban and urban systems are more common in Bamako District (Fig 1).

## Seroprevalence of brucellosis in small ruminants

Sensitivity and specificity calculations of the combination in parallel of Rose Bengal and cELISA gave a sensitivity (Se = 99.6%) and specificity (Sp = 97.3%). Table 2 shows the seroprevalence of brucellosis according to animal species, sex, age category, region and herd size. Overall, the seroprevalence was 4.1% [2.8–5.6 (95% CI)] and 25.2% [17.7–33.9 (95% CI)] for individual and herd, respectively. Sheep had higher seroprevalence than goats, but the difference was not significant (4.6%, [1.5–5.6 (95% CI)] *vs* 3.1% [3.0–6.8 (95% CI)]). Regarding animal sex, females had higher seroprevalence than males; however, the difference was not significant (4.3% [2.8–6.1 (95% CI)] *vs* 3.3% [1.3–6.6 (95% CI)]). The seroprevalence in young animals was 2.9% [1.0–6.3 (95% CI)] and that of the oldest was 4.8% [2.2–9.0 (95% CI)]. There was variation in seroprevalence according to localities (villages, municipalities, regions), however, the highest seroprevalence was found in the region of Segou [6.3; 3.7–9.7 (95% CI)]

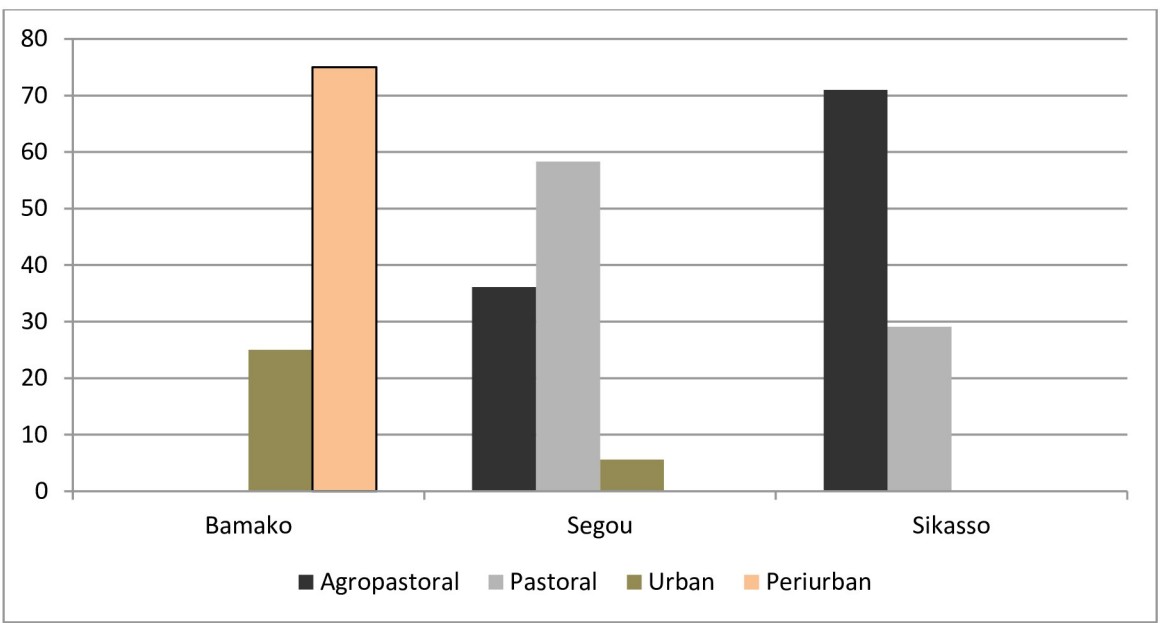

**Fig 1. Proportion of husbandry systems identified according to regions.**

compared to Bamako [4.2; 2.0–7.7 (95% CI)] and Sikasso [2.0; 0.8–4.1 (95% CI)]. However, the seroprevalence did not differ according the size of the herd (Table 2).

## Seroprevalence according to the husbandry systems

Seroprevalence varied between the different farming systems identified. Farms located in the peri-urban zone were the most affected compared to other husbandry systems. However, this difference was not significant (Table 3).

## Risky behaviors of farmers for the transmission of brucellosis in animals

Behaviors such as exchange of reproductive bull (30.2%), hanging of placentas (31.1%) in the farms, keeping in the herds females with history of abortion (69.7%) and mixing grazing herds (56%) were identified as risky behaviours that could foster transmission between animals. Close and prolonged contact (51.2%) with animals, consumption of unpasteurized dairy products (26.9%), assisting female animals during delivery without any protection (40.3%) were practices identified as risky for human to be infected by *Brucella* spp. According to the husbandry systems, there was a great variability in risk behaviors. Risky behaviours were observed more among pastoralists in pastoral areas as compared to other husbandry systems.

## Risk factors for animal contamination

Intrinsic factors such as species, age, sex, breed and husbandry systems have not been identified as risk factors influencing the seropositivity of animals. It was observed that herds of more than one hundred (100) animals were more likely to get brucellosis. Animals living in the municipalities of Katiena and Cinzana (all in the region of Segou) were more likely to be infected. Structurally, the lack of enclosures predispose animals to brucellosis. Occurrence of risky behaviors in a certain locality would favor the contamination of animals (Table 4).

**Table 2. Univariate association between brucellosis positivity and potential risk factors in goats and sheep (CI: Confidence interval).**

| Variables | Number tested | Seroprevalence (%) | CI 95 (%) |
|---|---|---|---|
| **Overall** | | | |
| Individual | 860 | 4.1 | 2.8–5.6 |
| Herd | 119 | 25.2 | 17.7–33.9 |
| **Species** | | | |
| Sheep | 514 | 4.6 | 1.5–5.6 |
| Goats | 346 | 3.1 | 3.0–6.8 |
| **Sex** | | | |
| Males | 212 | 3.3 | 1.3–6.6 |
| Females | 648 | 4.3 | 2.8–6.1 |
| **Age categories (months)** | | | |
| [6–17] | 202 | 2.9 | 1.0–6.3 |
| [18–29] | 237 | 4.6 | 2.3–8.1 |
| [30–41] | 236 | 3.8 | 1.7–7.1 |
| [42–112] | 185 | 4.8 | 2.2–9.0 |
| **Regions** | | | |
| Bamako | 234 | 4.2 | 2.0–7.7 |
| Segou | 285 | 6.3 | 3.7–9.7 |
| Sikasso | 341 | 2.0 | 0.8–4.1 |
| **Herd size** | | | |
| [10–20] | 312 | 4.8 | 2.7–7.8 |
| [21–40] | 317 | 3.7 | 1.9–6.5 |
| [41–60] | 96 | 0 | 0.0–0.3 |
| [61–80] | 22 | 9.1 | 1.1–29.1 |
| [81–100] | 41 | 2.4 | 0.0–12.8 |
| [over 101] | 72 | 6.9 | 2.2–15.4 |

Data stem from data collected from November 5 to December 12, 2018 in the cercle of Bamako, Segou and Sikasso, (%) = percentage, CI: Confidence interval.

## Discussion

This study is one of the few studies conducted in small ruminants in Mali to assess the magnitude and risks for transmission of brucellosis. Most of the studies regarding brucellosis in the country focussed on cattle [17,21,25]. The sera collected were subjected to Rose Bengal test and Competitive ELISA in parallel. The individual overall seroprevalence with the RBT was 2.2%, whilst with cELISA was 2.1%. The combination in parallel of Rose Bengal and ELISA gave a sensitivity of 99.6% and specificity of 97.3%. These values are slightly higher than those obtained by Adamou (2014) in Niger (Se = 97.28%; Sp = 93.57%) [26]. In this study, the

**Table 3. Individual and herd seroprevalence according to husbandry systems.** Data stem from study conduct in three regions of Mali.

| Husbandry systems | Number tested | Individual seroprevalence (%) | CI 95 (%) | Number herd tested | Herd seroprevalence (%) | CI 95 (%) |
|---|---|---|---|---|---|---|
| Agropastoral | 338 | 3.5 | 1.8–6.1 | 52 | 21.1 | 11.1–34.7 |
| Pastoral | 276 | 4.3 | 2.3–7.8 | 37 | 24.3 | 11.7–41.2 |
| Peri urban | 191 | 4.7 | 2.1–8.7 | 21 | 38.1 | 18.1–61.5 |
| Urban | 55 | 3.6 | 0.4–12.5 | 9 | 22.2 | 28.1–60.1 |

Data stem from data collected from November 5[th] to December 12[th], 2018 in the regions of Bamako, Segou and Sikasso, (%) = percentage, CI = Confidence interval.

**Table 4. Risk factors for animal contamination.**

| Factors | Odds Ratio | *p*\* | (95% C I) |
|---|---|---|---|
| No enclosure | 3.2 | 0.01 | 1,3–7.9 |
| Placentas on farm | 4.3 | 0.03 | 1.0–17.7 |
| Keep animals with hygromas | 2.7 | 0.02 | 1.0–6.2 |
| Herd size >100 | 5.7 | 0.01 | 1.4–2.3 |
| Municipality of Katiena | 9 | 0.009 | 1.9–64.5 |
| Municipality of Cinzana | 6.6 | 0.04 | 1.0–53.5 |
| Bamako Region | 3.8 | 0.02 | 1.2–12.9 |
| Segou Region | 5.8 | 0.001 | 2.0–18.3 |

Data stem from data collected from November 5th to December 12th, 2018 in the cercle of Bamako, Segou and Sikasso, P-value based on; P-value based on $\chi^2$-test or Fisher's exact test; and with statistically significant p<0.05.

individual seroprevalence obtained was 4.1%, which is higher than what Sow et *al.* (2015) [20] who obtained 0.5% in the municipality of Cinzana and lower than what was obtained by Coulibaly (2016) (37.1%) [21] in the circle of Niono all from the region of Segou in Mali. This seroprevalence is different from the results from Boukary et *al.*, (2013) (2.6%) [12] in Niger as well as those of Kanouté et al, 2017 (0%) [29] in Côte d'Ivoire and Dean et al., (2013) [30] in Togo (0.0%). This result is also different from seroprevalences of 0.5% and 11.4% obtained in Chad and Sudan respectively [15,31]. The present results showed a statistical difference in seroprevalences by region (p<0.05). This result could be due to the climatic characteristics of these zones as well as the farming methods practiced. Indeed, according to Akakpo (1987) [32], hot and humid areas would favour an increase of the seroprevalence of brucellosis. He also argued that a high animal concentration in an area would favour animal infection. This was somewhat confirmed by our results. The Segou region, which is the most affected, is a highly humid area irrigated by the Niger River, in addition to having the largest number of small ruminants among the three regions investigated. However, the seroprevalence found in the Sikasso area was not in line with the findings of Akakpo (1987) [32]. Indeed, the Sikasso region which is the wettest area in Mali had the lowest seroprevalence. Statistics show that living in the Sikasso region is a protective factor for an animal. This findingcould probably be attributed to livestock systems in the area. Indeed, the study showed that peri-urban livestock systems are the most affected (38.1%) followed by pastoral systems (24.3%). However, statistical analyses showed that these differences are not significant (p>0.05). These results corroborate those found in Ethiopia and Sudan [14,15] where the husbandry systems have no influence on the animal seropositivity. They differ from many studies that noted the influence of livestock systems on the seropositivity of animals. Thus, Mai et *al.*, (2012) [13] highlighted the role of pastoral systems whilst in Niger, Boukary et *al.*, 2013 [12] found that the seroprevalence of brucellosis is higher in urban livestock systems followed by peri-urban systems [33].

The difference in the conclusions from various authors may be due to animal intrinsic (age, sex, species) and/or extrinsic (herd size, site, husbandry systems) factors [32]. The differences in seroprevalence observed between our results and those obtained in other studies could also be explained by the use of different types of methodology (study design, sampling, study areas, species) [12,32,34]. Also, different types of tests with different sensitivities and specificities and also interpretations that often varied. Indeed, various serological tests are used in the diagnosis of brucellosis by various authors. Among these tests are RBT, cELISA, iELISA and Sero Agglutination of Wright (SAW). These tests have different specificities and sensitivities. These differences can influence the results obtained. [12,35,36].

The study revealed that farmers and/or animal owners have risky behaviors that could foster the transmission of brucellosis to both animals and humans. The observation of risk behaviours such as direct contact with animals and the consumption of raw milk corroborate those found by Steinmann et al (2006) in febrile patients in the periphery of the district of Bamako [25], those observed by Dao et al (2009) in febrile patients in Mopti [37]. The same observations were made by Tialla et al (2014) in the periurban farms of Dakar [24]. Risky behaviours such as cohabitation with animals have been reported by Osoro et al, (2015) in Kenya. They demonstrate that cohabitation with goats increases the seroprevalence of humans by six fold compared to those who do not live with animals [11].

This study is limited by the fact that no blood was collected from humans to see possible links between animal and human seropositivity. Also, in small ruminants the study was limited to determining the seroprevalence. Seropositivity indicates exposure but does not explain duration of an infection. The finding of antibodies in a single serum sample only indicates that infection has occurred sometime in the past which make its value as indicator of active infection limited. Furthermore, we concluded that *Brucella* species are circulating in small ruminants in Mali but we are unable to say which *Brucella* species as serological tests are not able to distinguish between the smooth *Brucella* spp. This would be possible by using bacteriological and or molecular tests.

## Conclusion

The present study confirmed the circulation of brucellosis in small ruminants in Mali. While the individual prevalence is low, the herd prevalence is relatively high and risky behaviors are observed in all types of husbandry systems. However, these risky behaviors expose mainly pastoralists (nomadic) in pastoral areas. Intervention strategies must be developed in order to reduce risky behaviors that could promote the transmission of brucellosis. These interventions can consist in raising farmers' awareness of the danger posed by brucellosis; training farmers on good farming practices to avoid contamination and / or the maintenance of brucellosis in the herd and train farmers on good hygiene practices to avoid contamination of humans. More studies are needed to identify the circulating *Brucella* strains both in animals and humans to characterize the links between animals and human infection for control strategies using a multi-sectoral and multidimensional approach.

## Supporting information

**S1 File.**
(PDF)

**S2 File.**
(PDF)

**S3 File.**
(PDF)

**S4 File.**
(PDF)

**S5 File.**
(RAR)

**S6 File.**
(XLSX)

**S7 File.**
(XLS)

## Author Contributions

**Conceptualization:** Souleymane Traoré, Richard B. Yapi, Kadiatou Coulibaly, Coletha Mathew, Gilbert Fokou, Rudovick R. Kazwala, Bassirou Bonfoh, Rianatou Bada Alambedji.

**Data curation:** Souleymane Traoré.

**Formal analysis:** Souleymane Traoré, Rianatou Bada Alambedji.

**Funding acquisition:** Rudovick R. Kazwala, Bassirou Bonfoh.

**Investigation:** Souleymane Traoré, Richard B. Yapi, Kadiatou Coulibaly.

**Methodology:** Souleymane Traoré, Richard B. Yapi, Kadiatou Coulibaly, Coletha Mathew, Gilbert Fokou, Rudovick R. Kazwala, Bassirou Bonfoh, Rianatou Bada Alambedji.

**Project administration:** Rudovick R. Kazwala, Bassirou Bonfoh.

**Resources:** Rudovick R. Kazwala, Bassirou Bonfoh.

**Supervision:** Kadiatou Coulibaly, Gilbert Fokou, Rudovick R. Kazwala, Bassirou Bonfoh, Rianatou Bada Alambedji.

**Validation:** Souleymane Traoré, Gilbert Fokou, Rudovick R. Kazwala, Bassirou Bonfoh, Rianatou Bada Alambedji.

**Visualization:** Souleymane Traoré, Coletha Mathew, Rianatou Bada Alambedji.

**Writing – original draft:** Souleymane Traoré.

**Writing – review & editing:** Souleymane Traoré, Richard B. Yapi, Kadiatou Coulibaly, Coletha Mathew, Gilbert Fokou, Rudovick R. Kazwala, Bassirou Bonfoh, Rianatou Bada Alambedji.

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
