## [Decision Letter · Decision Letter 0]

24 Sep 2020

PONE-D-20-21180

Seroprevalence of brucellosis in small ruminants and associated risk behaviours for humans in different husbandry systems in Mali

PLOS ONE

Dear Dr. Souleymane,

Thank you for submitting your manuscript to PLOS ONE. After careful consideration, we feel that it has merit but does not fully meet PLOS ONE’s publication criteria as it currently stands. Therefore, we invite you to submit a revised version of the manuscript that addresses the points raised during the review process.

We look forward to receiving your revised manuscript.

Kind regards,

Jasbir Singh Bedi

Academic Editor

PLOS ONE

Journal Requirements:

2. Please address the following:

- Please confirm that parental consent was not only requested, but obtained for participants who were minors.

- Please include additional information regarding the survey or questionnaire used in the study and ensure that you have provided sufficient details that others could replicate the analyses. For instance, if you developed a questionnaire as part of this study and it is not under a copyright more restrictive than CC-BY, please include a copy, in both the original language and English, as Supporting Information.

- In your Methods section, please provide additional information about the demographic details of your participants. Please ensure you have provided sufficient details to replicate the analyses such as: a)  a description of any inclusion/exclusion criteria that were applied to participant inclusion in the analysis, b) a table of relevant demographic details, c) a statement as to whether your sample can be considered representative of a larger population.

- Please ensure you have thoroughly discussed any potential limitations of this study within the Discussion section.

- Please state the dates during which data collection took place.

3.We note that you have indicated that data from this study are available upon request. PLOS only allows data to be available upon request if there are legal or ethical restrictions on sharing data publicly. For information on unacceptable data access restrictions, please see http://journals.plos.org/plosone/s/data-availability#loc-unacceptable-data-access-restrictions.

4.We note that [Figure(s) 1] in your submission contain [map/satellite] images which may be copyrighted. All PLOS content is published under the Creative Commons Attribution License (CC BY 4.0), which means that the manuscript, images, and Supporting Information files will be freely available online, and any third party is permitted to access, download, copy, distribute, and use these materials in any way, even commercially, with proper attribution. For these reasons, we cannot publish previously copyrighted maps or satellite images created using proprietary data, such as Google software (Google Maps, Street View, and Earth). For more information, see our copyright guidelines: http://journals.plos.org/plosone/s/licenses-and-copyright.

1.    You may seek permission from the original copyright holder of Figure(s) [1] to publish the content specifically under the CC BY 4.0 license. 

5. We note you have included a table to which you do not refer in the text of your manuscript. Please ensure that you refer to Table 2 in your text; if accepted, production will need this reference to link the reader to the Table.

6. Please include a copy of Table 3 which you refer to in your text on page 13.

Reviewers' comments:

Reviewer's Responses to Questions

**Comments to the Author**

1. Is the manuscript technically sound, and do the data support the conclusions?

Reviewer #1: Partly

2. Has the statistical analysis been performed appropriately and rigorously? 

Reviewer #1: No

3. Have the authors made all data underlying the findings in their manuscript fully available?

Reviewer #1: Yes

4. Is the manuscript presented in an intelligible fashion and written in standard English?

Reviewer #1: Yes

5. Review Comments to the Author

Reviewer #1: Manuscript Number: PONE-D-20-21180

Title: “Seroprevalence of brucellosis in small ruminants and associated risk behaviours for humans in different husbandry systems in Mali”

I appreciate the efforts by the authors to address the role of small ruminant husbandry systems in maintaining and transmitting brucellosis in Mali. Brucellosis remain an important endemic zoonoses in many of the developing nations, where such epidemiological studies need to be carried out to reveal the true prevalence and risk factors for the disease. I have major concern with the presentation of the results in terms of possible risk factors without the calculation of the measures of association, which I have highlighted in comment number 4.

Comments:

1. Line 53-55: “Livestock in Mali is facing many health concerns and many diseases hinder the development of the production in the country among which brucellosis [3]”…. The sentence seems to be incomplete, rephrase it.

2. Line 56: “The species B. melitensis which is the most widespread in small ruminants seems to be the most pathogenic in humans”: Provide the reference for it.

3. Line 106: “In each locality (village, hamlet), six farms were chosen”: The rationale for choosing six farms need to be described.

4. Results: The major issue with results are that there is no calculation for measures of association, the authors depicted the results just in % seroprevalence for different parameters. The univariate and multivariate analysis of the possible risk factors need to be estimated to justify the objectives of the study in terms of risk factors.

5. Line 234: “Risk behaviors of brucellosis transmission identified among farmers”: The authors presented the frequency of the possible risk factors for brucellosis transmission on the basis of the responses to the questionnaire. There is also need to find the association with the farm level prevalence of brucellosis. The title is little misnomer, the authors didn’t study the transmission pathways of brucellosis and there is no supporting data for human brucellosis prevalence.

6. Line 247: “this is an up-to-date study conducted”: The sentence needs to be revised.

7. Line 249: We used two diagnostic “technics”: Spell check it.

8. Line 252: “Seroprevalence and risk behaviors will be the main points around which the discussion will be structured”.: Need to be revised.

9. Line 265-267: “He also argued that a high animal concentration in an area would favour animal infection. This was somewhat confirmed by our results.”: There is simply the mention of herd size and seroprevalence in Table 1, the association between these need to be estimated before arriving any conclusion.

10. Figure 2: In Segou, the bar chart needs to be revised to bring urban bar together with rest of the two

11. Figure 3: Brucellosis risk behaviors identified according to small ruminant’s husbandry systems: I don’t think that this figure provides any significant finding, it can be supplementary Table.

6. PLOS authors have the option to publish the peer review history of their article (what does this mean?). If published, this will include your full peer review and any attached files.

Reviewer #1: No

---

## [Author Response · Author response to Decision Letter 0]

2 Dec 2020

#Responses

• Please address the following: - Please confirm that parental consent was not only requested, but obtained for participants who were minors.

#####The consent of each minor interviewed was obtained in addition to that of the responsible adult (see consent form attached). The statement has been changed to reflect that the consent for participants who were minors was actually obtained Line [205] 

• Please include additional information regarding the survey or questionnaire used in the study and ensure that you have provided sufficient details that others could replicate the analyses. For instance, if you developed a questionnaire as part of this study and it is not under a copyright more restrictive than CC-BY, please include a copy, in both the original language and English, as Supporting Information.

 #####English and French versions of the questionnaire have been attached (see the attached file)

• In your Methods section, please provide additional information about the demographic details of your participants. Please ensure you have provided sufficient details to replicate the analyses such as: a) a description of any inclusion/exclusion criteria that were applied to participant inclusion in the analysis, b) a table of relevant demographic details, c) a statement as to whether your sample can be considered representative of a larger population.

##########

a. Inclusion / exclusion criteria are describe in the methods section line [98 - 101]

b. Demographic details: Demographic details about study participants have been added in the results section. line [204 - 213]

c. Sample representativeness: The sample can be confidently considered as representative of a larger population because we used random sampling method at all levels.

• Please ensure you have thoroughly discussed any potential limitations of this study within the Discussion section.

#####The limitations of the study are discussed in the discussion section. Line [331 - 339]

• Please state the dates during which data collection took place.

Month and dates of the study are include in the ‘’methods section’’ line [75]

• We note that you have indicated that data from this study are available upon request. PLOS only allows data to be available upon request if there are legal or ethical restrictions on sharing data publicly. For information on unacceptable data access restrictions, please see http://journals.plos.org/plosone/s/data-availability#loc-unacceptable-data-access-restrictions.

#####The data are attached with the files associated with the manuscript

Reviewer #1: Manuscript Number: PONE-D-20-21180 Title: “Seroprevalence of brucellosis in small ruminants and associated risk behaviours for humans in different husbandry systems in Mali” I appreciate the efforts by the authors to address the role of small ruminant husbandry systems in maintaining and transmitting brucellosis in Mali. Brucellosis remain an important endemic zoonoses in many of the developing nations, where such epidemiological studies need to be carried out to reveal the true prevalence and risk factors for the disease. I have major concern with the presentation of the results in terms of possible risk factors without the calculation of the measures of association, which I have highlighted in comment number 4. 

Comments: 

1. Line 53-55: “Livestock in Mali is facing many health concerns and many diseases hinder the development of the production in the country among which brucellosis [3]”…. The sentence seems to be incomplete, rephrase it

#####The sentence has been rephrased, line [54 - 55]

2. Line 56: “The species B. melitensis which is the most widespread in small ruminants seems to be the most pathogenic in humans”: Provide the reference for it. 

#####The references have been added, line [59]

3. Line 106: “In each locality (village, hamlet), six farms were chosen”: The rationale for choosing six farms need to be described. 

#####The rationale has been described Line [106 - 110]

4. Results: The major issue with results are that there is no calculation for measures of association, the authors depicted the results just in % seroprevalence for different parameters. The univariate and multivariate analysis of the possible risk factors need to be estimated to justify the objectives of the study in terms of risk factors. 

We performed logistic regression analysis to determine association between the predetermined risk factors and seropositivity. See in the methodology section Line [178 - 183], In the results section line [267 - 278] and in the discussion section.

5. Line 234: “Risk behaviors of brucellosis transmission identified among farmers”: The authors presented the frequency of the possible risk factors for brucellosis transmission on the basis of the responses to the questionnaire. There is also need to find the association with the farm level prevalence of brucellosis. The title is little misnomer, the authors didn’t study the transmission pathways of brucellosis and there is no supporting data for human brucellosis prevalence 

#####In the present study we focussed on the seroprevalence of brucellosis in animals and risk behaviours that would facilitate transmission of the disease/infection within animals and between animals and humans. 

One of the limitations of the study is the fact that we did not perform serological tests in humans. The study was limited to the observation of presence of risky behaviours as identified by several authors. Presence of infected animals and risky behaviours are the critical as will facilitate transmission of the disease. The next stage is now to check on the exposure status on the humans. 

6. Line 247: “this is an up-to-date study conducted”: The sentence needs to be revised 

#####[the sentence has been revised line 280 - 282].

7. Line 249: We used two diagnostic “technics”: Spell check it 

##########(the sentence has been deleted

8. Line 252: “Seroprevalence and risk behaviors will be the main points around which the discussion will be structured”: Need to be revised. 

##########This sentence has been deleted

9. Line 265-267: “He also argued that a high animal concentration in an area would favour animal infection. This was somewhat confirmed by our results.”: There is simply the mention of herd size and seroprevalence in Table 1, the association between these need to be estimated before arriving any conclusion 

##########We performed a regression analysis and we found the association between herd size and seropositivity [see table 4]. 

10. Figure 2 (now Figure 1): In Segou, the bar chart needs to be revised to bring urban bar together with rest of the two 

##########The urban bar has been brought closer to the other two bars See Figure 1. 

11. Figure 3: Brucellosis risk behaviors identified according to small ruminant’s husbandry systems: I don’t think that this figure provides any significant finding, it can be supplementary Table.

##########We have removed figure 3 as most of the information is found in the text.

---

## [Editor Report · Decision Letter 1]

9 Dec 2020

PONE-D-20-21180R1

Seroprevalence of brucellosis in small ruminants and related risk behaviours among humans in different husbandry systems in Mali

PLOS ONE

Dear Dr. Souleymane ,

Thank you for submitting your manuscript to PLOS ONE. After careful consideration, we feel that it has merit but does not fully meet PLOS ONE’s publication criteria as it currently stands. Therefore, we invite you to submit a revised version of the manuscript that addresses the points raised during the review process.

The manuscript needs minor revision in the form of spelling checks and typo errors. Please check it carefully and the re-submit.

We look forward to receiving your revised manuscript.

Kind regards,

Jasbir Singh Bedi

Academic Editor

PLOS ONE

---

## [Author Response · Author response to Decision Letter 1]

23 Dec 2020

##### Responses to reviewers

The whole document has been checked, spelling and typographical errors have been revised.

---

## [Editor Report · Decision Letter 2]

26 Dec 2020

Seroprevalence of brucellosis in small ruminants and related risk behaviours among humans in different husbandry systems in Mali

PONE-D-20-21180R2

Dear Dr. Souleymane,

We’re pleased to inform you that your manuscript has been judged scientifically suitable for publication and will be formally accepted for publication once it meets all outstanding technical requirements.

Kind regards,

Jasbir Singh Bedi

Academic Editor

PLOS ONE
---

## [Editor Report · Acceptance letter]

2 Jan 2021

PONE-D-20-21180R2 

Seroprevalence of brucellosis in small ruminants and related risk behaviours among humans in different husbandry systems in Mali 

Dear Dr. Traoré:

I'm pleased to inform you that your manuscript has been deemed suitable for publication in PLOS ONE. Congratulations! Your manuscript is now with our production department. 

Kind regards, 

on behalf of

Dr. Jasbir Singh Bedi 

Academic Editor

PLOS ONE